EMBO
reports

# Post-metaphase correction of aberrant kinetochore-microtubule attachments in mammalian eggs

Anna Kouznetsova[1],* , Tomoya S Kitajima[2], Hjalmar Brismar[3] & Christer Höög[1]

## Abstract

The accuracy of the two sequential meiotic divisions in oocytes is essential for creating a haploid gamete with a normal chromosomal content. Here, we have analysed the 3D dynamics of chromosomes during the second meiotic division in live mouse oocytes. We find that chromosomes form stable kinetochore–microtubule attachments at the end of prometaphase II stage that are retained until anaphase II onset. Remarkably, we observe that more than 20% of the kinetochore–microtubule attachments at the metaphase II stage are merotelic or lateral. However, < 1% of all chromosomes at onset of anaphase II are found to lag at the spindle equator and < 10% of the laggards missegregate and give rise to aneuploid gametes. Our results demonstrate that aberrant kinetochore–microtubule attachments are not corrected at the metaphase stage of the second meiotic division. Thus, the accuracy of the chromosome segregation process in mouse oocytes during meiosis II is ensured by an efficient correction process acting at the anaphase stage.

**Keywords** aneuploidy; meiosis; oocyte; second meiotic division; segregation
**Subject Categories** Cell Cycle; Development & Differentiation

## Introduction

Partitioning of chromosomes during cell division in mammals takes place when sister chromatids (mitosis/meiosis II) or homologous chromosomes (meiosis I) undergo a symmetrical segregation process at the anaphase stage. This results in formation of diploid (mitosis/meiosis I) or haploid (meiosis II) daughter cells with a balanced chromosomal content. Missegregation of chromosomes during cell division gives rise to aneuploidy in daughter cells, genetic abnormalities that could contribute to tumour development (mitosis) or birth defects and infertility (meiosis) [1].

Cell divisions in somatic cells and germ cells show low missegregation rates typically not exceeding 4–5%; for instance, < 2% of the cell divisions in oocytes in young mice result in embryos with an abnormal amount of chromosomes [2]. In sharp contrast to this, 20–25% of the oocytes in women at an age of 30 are found to be aneuploid after the second meiotic division, and the aneuploidy rate further increases with age, a phenomenon that has a considerable impact on the health of the human population [3,4]. The observed aneuploidies in human eggs originate from both meiosis I and meiosis II cell divisions, with approximately half of the meiosis II missegregation events arising independently from meiosis I errors [5,6]. The reasons for the high aneuploidy rate at the two meiotic divisions in human oocytes are not known.

The accuracy of the chromosome segregation process is ensured by bi-orientation of chromosomes (mitosis/meiosis II) and homologous chromosomes (meiosis I) at the spindle equator. The bi-orientation process requires that the kinetochore [7], a macromolecular protein structure bound to the centromere of each chromosome, separately (in mitosis/meiosis II) or as a single entity (meiosis I) is attached to microtubules (MTs) from the spindle poles. Bi-directionally, end-on attached chromosomes (amphitelic attachments) give rise to symmetrical tension across the spindle equator and satisfy the spindle assembly checkpoint (SAC), promoting anaphase onset [8]. Erroneous kinetochore–MT attachments are frequently observed during mitosis, including merotelic attachments (where a kinetochore is attached to MTs from opposite spindle poles) and syntelic attachments (where the sister kinetochores of one chromosome are attached to MTs from the same spindle pole) [9,10], whereas merotelic attachments are seen during meiosis I (MI) [11,12] and meiosis II (MII) [13]. Kinetochores can also attach to the lateral surfaces of microtubules forming so-called lateral attachments that act as important intermediates during bi-orientation of chromosomes [14]. In situations of lateral, merotelic or syntelic attachments, the force balance between sister kinetochores becomes unevenly distributed. Reduced inter-kinetochore or intra-kinetochore tension at the spindle equator is sensed by an error correction pathway, involving the Aurora family of kinases (Aurora B during mitosis and Aurora B/C during meiosis) [15,16]. Aurora

1 Department of Cell and Molecular Biology, Karolinska Institutet, Stockholm, Sweden
2 Laboratory for Chromosome Segregation, RIKEN Center for Biosystems Dynamics Research, Kobe, Japan
3 Science for Life Laboratory, Department of Applied Physics, Royal Institute of Technology, Solna, Sweden
*Corresponding author. Tel: +46(8)52487159; E-mail: anna.kouznetsova@ki.se

kinases are located at the inter-centromeric domain between sister kinetochores and destabilize kinetochore–MT attachments by phosphorylation of kinetochore-located substrates.

The second meiotic division in mammalian oocytes represents a unique cell division process with similarities and differences compared to mitosis and MI. Similar to mitosis, chromosome bi-orientation results in an equational segregation of chromatids to the opposite spindle poles at anaphase onset. MII oocytes, however, show cell division asymmetry, a larger cytoplasm volume and absence of centrosomes, features not shared with somatic cells. While these latter features, on the other hand, are shared with MI oocytes, the presence of homologous chromosomes in MI oocytes where the fused sister kinetochores of chromosomes form syntelic attachments differs from what is observed in both MII oocytes and mitotic cells. Furthermore, distinct from both mitotic cells and MI oocytes, mouse and human MII oocytes are arrested at the metaphase stage and resume cell division only after fertilization.

Here, we have in a comprehensive manner examined the *in vivo* dynamics of chromosomes and centromeres from anaphase I to anaphase II in mouse oocytes using a high-resolution imaging procedure, followed by quantitative analysis of chromosome behaviour. The spatiotemporal profiles for all chromosomes in individual oocytes were determined, including assembly of chromosomes at the spindle equator at metaphase II and segregation of sister chromatids at anaphase II. We find that stable kinetochore–MT attachments are formed at the end of the prometaphase II stage, including also multiple aberrant forms of non-amphitelic attachments. These attachments are retained throughout the metaphase II stage, but unexpectedly do not result in aneuploidy in the resulting haploid gametes.

## Results

### The anaphase I to the metaphase II arrest period in mouse oocytes

The second meiotic division in mouse and human oocytes is divided in two phases by cytostatic factor (CSF)-dependent metaphase arrest [17]. We first analysed the transition period from anaphase onset of the first meiotic division (anaphase I) to cytostatic factor (CSF)-dependent arrest at metaphase stage of the second meiotic division (metaphase II).

At anaphase I, the bivalent chromosomes segregate into two daughter cells, a secondary oocyte and a deteriorating first polar body. The secondary oocyte has a diploid chromosome content where each of the 20 chromosomes consists of two chromatids held together at the centromere region. We used an H2B-mCherry fusion protein to visualize chromosomes and a CENP-C-EGFP fusion protein to label centromeres. The positioning of the centromeres was used to approximate kinetochore locations, as described previously by [Ref. 18]. Chromosome dynamics was documented every 5 min by time-lapse microscopy in secondary oocytes derived from young mice (10–13 weeks old) with a normal euploid karyotype (Fig 1 and Movie EV1). Transition from anaphase onset in MI oocytes until interkinesis required $40 \pm 7$ min to complete, whereas interkinesis, a period characterized by partial chromosome decondensation, lasted for another $40 \pm 7$ min (Fig EV1A). Onset of

prometaphase of the second meiotic division (prometaphase II) was defined by individualization of chromosomes and chromosome congression, a process that required $70 \pm 20$ min to be completed (Fig EV1A). Chromosome congression progressed considerably faster at the second meiotic division than reported for the first meiotic division (about 4 h, [18], but much slower than reported for human and mouse somatic cells (about 15 min [19,20]).

To get a more detailed understanding of chromosome dynamics from onset of anaphase I to metaphase II arrest, we tracked all 40 centromeres of the 20 chromosomes in space and time in 6 oocytes and followed chromosome positions (defined here as the midpoint between sister centromeres) relative to the centre (Fig 2A, see details in Materials and Methods). Chromosomes were maximally congregated at interkinesis (Figs 1 and 2A and EV1B). Entry into prometaphase in MII oocytes was characterized by intensive chromosome movements in parallel with each other towards the spindle poles (Figs 1 and 2B and EV1C and D). Most chromosomes relocated 6–7 μm from the centre (Fig 2A), whereas a few chromosomes moved up to 15 μm from the centre (dashed line in Fig 2A), approximately corresponding to the distance to the spindle pole (Fig EV2A). We did not observe the presence of an "equatorial ring/belt" at prometaphase II, a chromosome arrangement described at the prometaphase stage in human mitotic cells (present 1.5–10 min after NEBD) and at the prometaphase stage in mouse MI oocytes (present 30–120 min after GVBD), suggested to facilitate capture of kinetochores by MTs and ensure bi-orientation of chromosomes [18,19]. The mean chromosome speed at prometaphase II reached 0.3 μm/min (black line in Figs 2B and EV1C), similar to the speed observed for chromosomes during invasion of the central space before formation of the metaphase plate in MI oocytes [18]. Distant chromosomes moved more rapidly (1 μm/min) (Fig 2B, dashed line). The inter-centromere distance was $0.5 \pm 0.1$ μm at the beginning of prometaphase II and then increased to $1.6 \pm 0.2$ μm at the metaphase II stage (Figs 2C and EV1E). Notably, increase in inter-centromere distance at the prometaphase stage in MII oocytes was coupled to attaining a parallel orientation to the spindle axis (Fig EV1F).

Direct visualization of MTs and chromosomes at an early stage of prometaphase II in fixed oocytes by high-resolution confocal microscopy revealed that chromosomes with syntelic attachments (when both sister kinetochores interact with MTs emanating from the same spindle pole) were positioned close to the spindle poles and showed small inter-centromere distances (Fig 2D and E). In contrast, chromosomes displaying amphitelic attachments (when the sister kinetochores attach to the MTs emanating from opposite spindle poles) were localized close to the spindle equator and showed large inter-centromere distances (Fig 2D and E). Finally, chromosomes with at least one bi-directionally attached kinetochore (termed here merotelic/lateral attachments) were scattered between the two spindle poles and exhibited variable inter-centromere distances. A comparative analysis of oocytes at prometaphase II and at metaphase II showed that the fraction of chromosomes with amphitelic end-on attachments increased from $20 \pm 10\%$ to $76 \pm 4\%$. Importantly, more than 20% of the chromosomes (4–6 chromosomes per oocyte) retained merotelic/lateral attachments at the metaphase II stage (Fig EV2B). The aberrant attachments present at the metaphase II stage were not corrected despite the presence of Aurora B kinase on the kinetochores and Aurora C

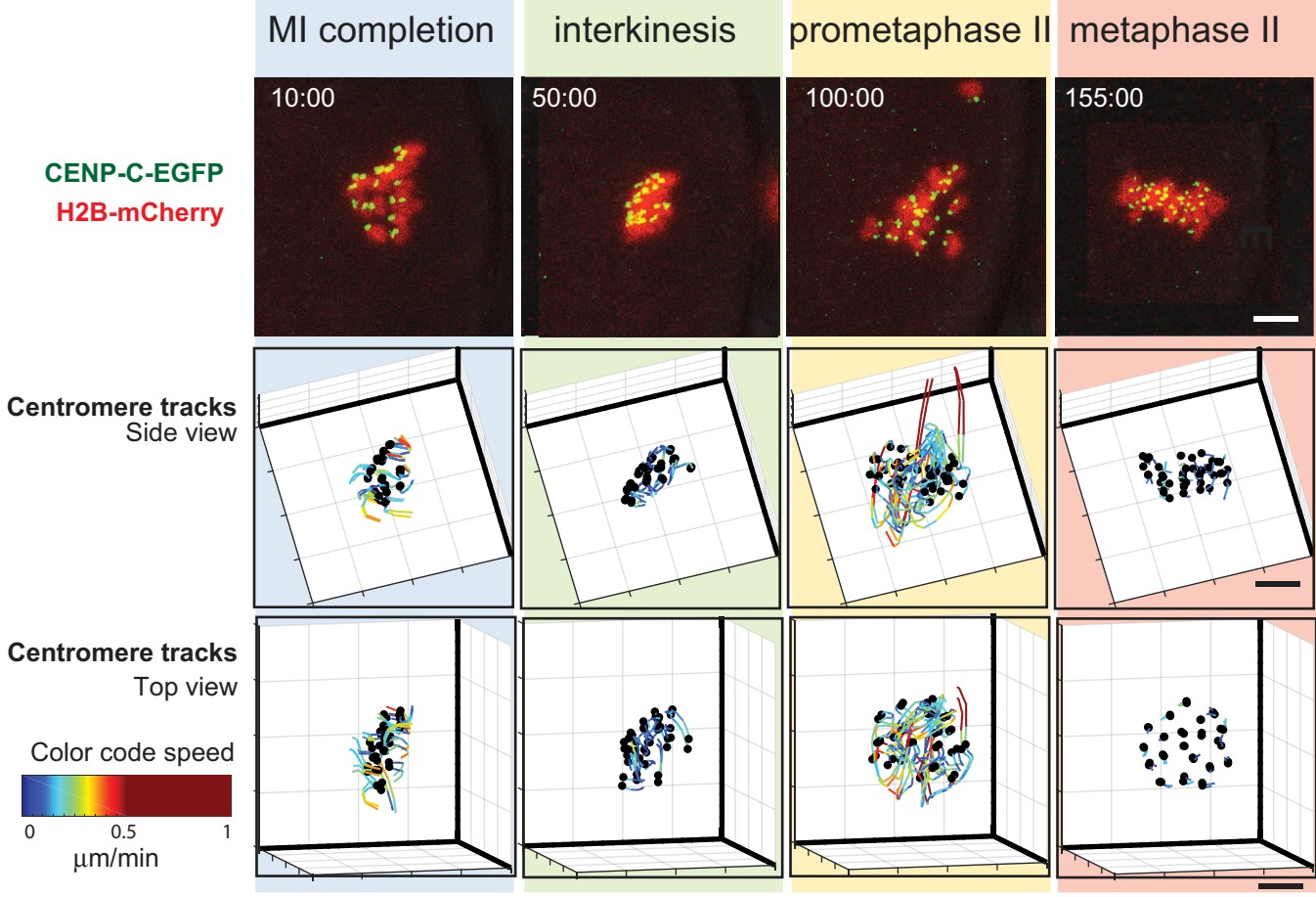

**Figure 1. 3D *in vivo* tracking of chromosome movements from anaphase I to metaphase II arrest.**

Time-lapse imaging of an oocyte expressing CENP-C-EGFP (centromeres, green) and H2B-mCherry (chromatin, red). Upper row shows maximum intensity z projection images from representative time points of four stages from anaphase I onset to CSF-dependent MII arrest. The 3D positions of centromeres are shown as black dots in a side view (along the spindle equator, middle row) and top view (perpendicular to the equator plane, bottom row). Tracks of individual centromeres are colour-coded according to the speed, as indicated by the colour bar. Time is shown after anaphase I onset (min:sec). Scale bars, 5 μm.

kinase at the inter-centromeric regions of prometaphase II and metaphase II chromosomes (Fig EV2C and D).

In summary, we show that chromosomes at the prometaphase stage in MII oocytes undergo rapid movements along parallel tracks towards the spindle poles, likely to be mediated by syntelic attachments. The chromosomes then bi-orient and become stably positioned at the spindle equator, a process taking place without the formation an intermediate equatorial ring structure. Surprisingly, approximately 20% of the chromosomes in oocytes retain merotelic/lateral attachments at the CSF-dependent metaphase II arrest stage.

**The metaphase to the anaphase transition in mouse MII oocytes**

We next studied chromosome behaviour in mouse oocytes upon release from CSF-mediated arrest. We labelled the centromeres of the H2B-mCherry-tagged chromosomes with CENP-C-EGFP, performed time-lapse imaging with 1.5- to 3-min time intervals following artificial activation and analysed the spatiotemporal behaviour of chromosomes in MII oocytes advancing from the metaphase to the anaphase stage of the second meiotic division.

A majority of the analysed MII oocytes displayed an error-free cell division process where the sister chromatids of the 20 chromosomes were observed to synchronously segregate to opposite spindle poles 40–140 min after activation (Fig 3 and Movie EV2). To get a detailed understanding of chromosome dynamics after release of the CSF-dependent arrest, we tracked all centromeres in space and time in 14 oocytes with synchronous chromatid separation and followed chromosome positions relative to the spindle equator plane and spindle axis (Fig 4A–C, see details in Materials and Methods). The chromosomes at the metaphase stage in MII oocytes upon release from CSF-mediated arrest attained a plate-like shape with a maximum distance to the spindle equatorial plane of $1.9 \pm 0.6$ μm and a maximum distance to the spindle axis of $4.6 \pm 0.6$ μm (Figs 4A and B, and EV3A). The metaphase chromosomes were oriented almost parallel to the spindle axis with a mean chromosome-axis angle of $6 \pm 2°$ (Figs 4C and EV3B and C), comparable to the chromosome-axis angle observed at the metaphase stage in MI oocytes [18], but slightly smaller than observed at the metaphase stage in human mitotic cells [19]. The inter-centromere distance for chromosomes at the metaphase II stage was

1.6 ± 0.2 μm, almost twice as large as what was observed at the metaphase stage in human and mouse mitotic cells [19,21] and did not change until anaphase II onset (Figs 4D and EV3D and E).

The mean chromosomes speed was as low as 0.1 ± 0.02 μm/min until anaphase II onset (Figs 3 and EV3F and G), resembling the average speed for chromosomes positioned at the metaphase plate

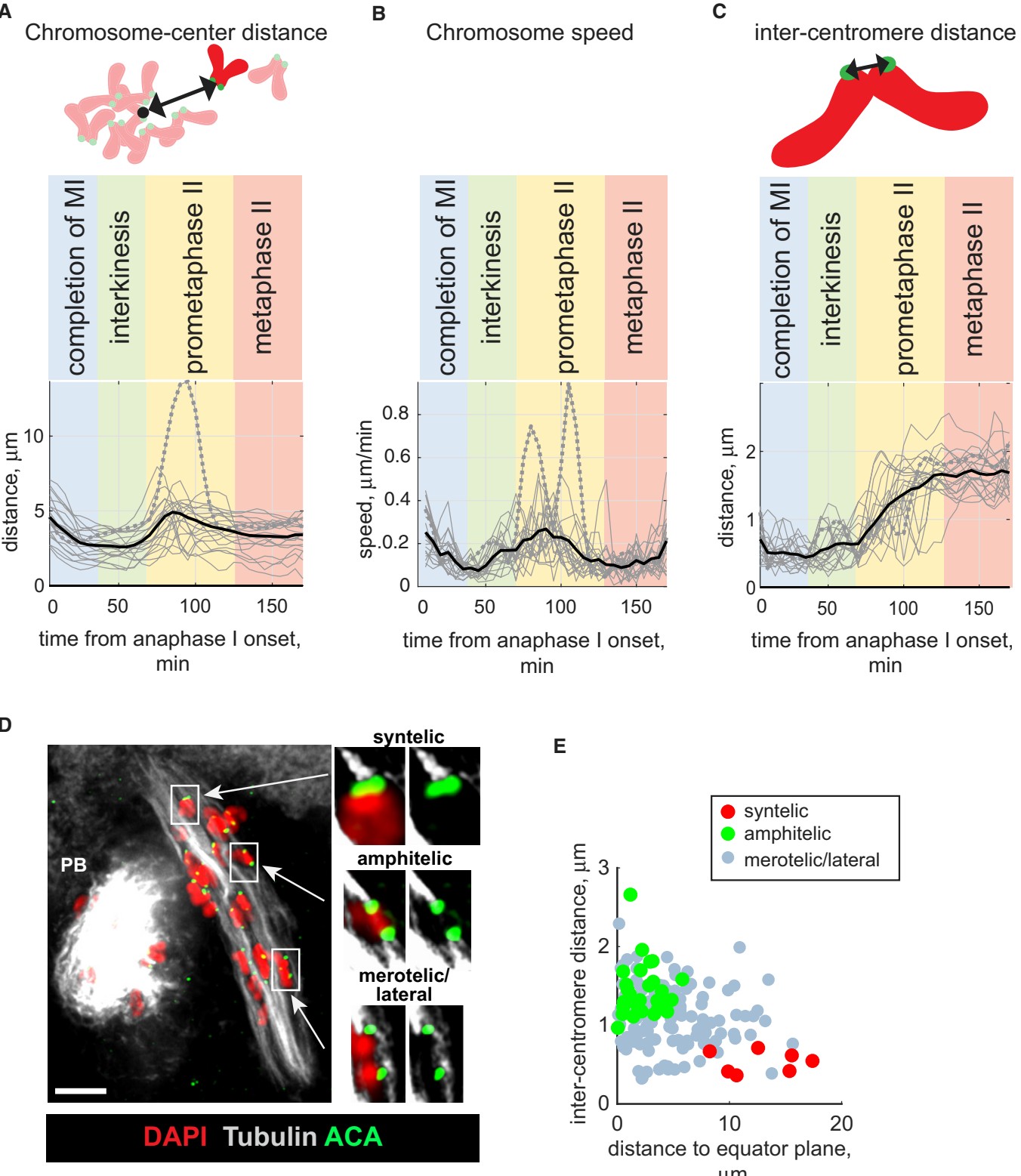

**Figure 2.**

**Figure 2. Chromosomes undergo rapid movements concurrent with inter-centromere stretching in prometaphase II.**

Chromosome parameters are shown in (A–C) on the vertical axis, and time after anaphase I onset is shown in min on the horizontal axis. Thin grey lines in (A–C) represent individual chromosomes; the distantly observed chromosome is highlighted by dashed line; the mean value for each time point is indicated by thick black line. The stages between anaphase I and metaphase II are colour-coded and labelled on the top of the charts. On the schemes above the charts in (A) and (C), chromosomes are red with green centromeres.

A   Changes in the distance to the centre for all chromosomes in a representative oocyte undergoing transition from anaphase I to metaphase II from Fig 1. The black arrow on the scheme above the chart indicates chromosome–centre distance.

B   Changes in the speed of all chromosomes observed for the oocyte from Fig 1.

C   Changes in inter-centromere distance (indicated on a scheme by the black arrow), observed for the oocyte from Fig 1.

D   Kinetochore–MT attachments at the prometaphase II stage were visualized using an anti-tubulin antibody (white), an anti-centromeric ACA antibody (green) and DAPI to label chromatin (red). The image represents maximum intensity projection through all z-planes containing MTs. Representative chromosomes displaying syntelic, amphitelic or merotelic/lateral attachments are enclosed in white frames, and their enlarged single z-plane images are shown to the right. PB: polar body. Scale bar, 10 μm.

E   Chromosomes with syntelic (red), amphitelic (green) and merotelic/lateral (grey) attachments were plotted according to their inter-centromere distance (on the horizontal axis) and distance to spindle equator (vertical axis). Data shown for 156 chromosomes taken from 8 prometaphase II oocytes.

in MI oocytes [18]. Thus, the position for chromosomes, their orientation, speed and inter-centromere distances were similar at the metaphase stage before and after CSF-mediated arrest release.

We next visualized kinetochore–MT attachments in fixed oocytes after release from CSF-mediated arrest and found that $78 \pm 4\%$ of the chromosomes displayed amphitelic attachments, while the remaining chromosomes (3–6 per oocyte) had bi-directional merotelic/lateral attachments (Figs 4E and F, and EV4A). This is more than previously reported (about 5% of attachments were scored as merotelic or lateral in MII oocytes by [Ref. 13]). The discrepancy between our result and those reported by [Ref. 13] is most probably explained by the use of a different visualization protocols. We have used a protocol described in [Ref. 16], where cells undergo cold treatment to remove less stable microtubules that are not attached to the kinetochores [22] in a stabilizing buffer instead of PBS as used by [Ref. 13], to preserve MTs. In addition, we acquired the images with a microscope equipped with an Airyscan microscope module (Zeiss) to achieve super-resolution, allowing improved identification of thin MTs. Importantly, the percentage of merotelic/lateral attachments that we score before and after the CSF-mediated metaphase arrest release did not change.

Inter-centromere distances for aberrantly attached chromosomes were reduced in comparison with chromosomes with amphitelic attachments (Fig 4F, $P = 0.001$, nested ANOVA), validating that merotelic/lateral attachments contribute to a reduced level of bi-directional tension. The aberrant merotelic/lateral attachments are retained despite the presence of Aurora B on the kinetochores and Aurora C at the inter-centromeric region of metaphase chromosomes after release of CSF-mediated arrest (Fig EV4B).

In summary, the behaviour of chromosomes and the nature of kinetochore–MT attachments at the metaphase II stage are the same before and after release of the CSF-mediated arrest. We conclude that stable kinetochore–MT attachments are formed at the end of the prometaphase stage and maintained until anaphase onset in MII oocytes. As a result of this, aberrant attachments formed at the prometaphase II stage, affecting more than 20% of the chromosomes in each oocyte at the MII stage, are not corrected prior to anaphase onset.

## The anaphase transition in MII oocytes

We have established that more than 20% of chromosomes at metaphase stage display merotelic or lateral attachments in MII oocytes.

Such aberrant attachments could give rise to laggards, chromatids that remain at the spindle midzone following anaphase onset and could contribute to aneuploidy. We monitored MII oocytes with H2B-mCherry-tagged chromosomes and CENP-C-EGFP-labelled centromeres from metaphase to anaphase by time-lapse imaging microscopy. Surprisingly, laggards at the anaphase stage were observed only in 15% of the imaged 71 oocytes. Thus, we find that most merotelic/lateral attachments affecting chromosomes at the metaphase II stage were not manifested as laggards at the anaphase II stage.

To better understand the behaviour and characteristics of the chromosomes that give rise to laggards at anaphase, we analysed 10 MII oocytes that contained 15 laggards at anaphase (Fig 5 and Movie EV3). The lagging chromatids did not delay anaphase II onset (Fig EV5A) and did not give rise to micronuclei, the latter in contrast to what have been observed in mitotic cells [1]. Centromere tracking revealed that the 15 lagging chromatids observed at anaphase II originated from 13 laggard-producing chromosomes. Most (9 out of 13) laggard-producing chromosomes were positioned among normally segregating chromosomes at the metaphase II plate prior to anaphase II onset (Fig 6A, normally segregating chromosomes are located between dashed lines). Likewise, there was no preferred position in relation to the spindle axis for the laggard-producing chromosomes (Fig 6B). The speed of laggard-producing chromosomes before anaphase II onset was low and similar to what was observed for chromosomes segregating without giving rise to laggards (Fig EV5B and C). However, the angle to the spindle axis for laggard-producing chromosomes was elevated to $17 \pm 10°$ ($6 \pm 2°$ for normally segregating chromosomes) and their mean inter-centromere distance reduced to $1.3 \pm 0.7$ μm at the metaphase II stage ($1.7 \pm 0.2$ μm for normally segregating chromosomes, $P < 0.001$, two-way ANOVA; Figs 6C and D, and EV5D and E). The mean inter-centromere distance was found to be negatively correlated with the mean chromosome-axis angle (Pearson's correlation coefficient is 0.6, $P = 0.03$; Fig EV5F). Furthermore, a reduction of the inter-centromere distance for laggard-producing chromosomes in most cases coincided with an increase in chromosome-axis angle (Fig EV5G), suggesting that a decrease in inter-centromere distance triggers chromosome rotation. Thus, chromosomes in MII oocytes that give rise to laggards at the anaphase stage have a stable position at the metaphase stage and show a reduced inter-centromere distance and an increased chromosome-axis angle. The reduced inter-centromere distance indicates that laggard-producing

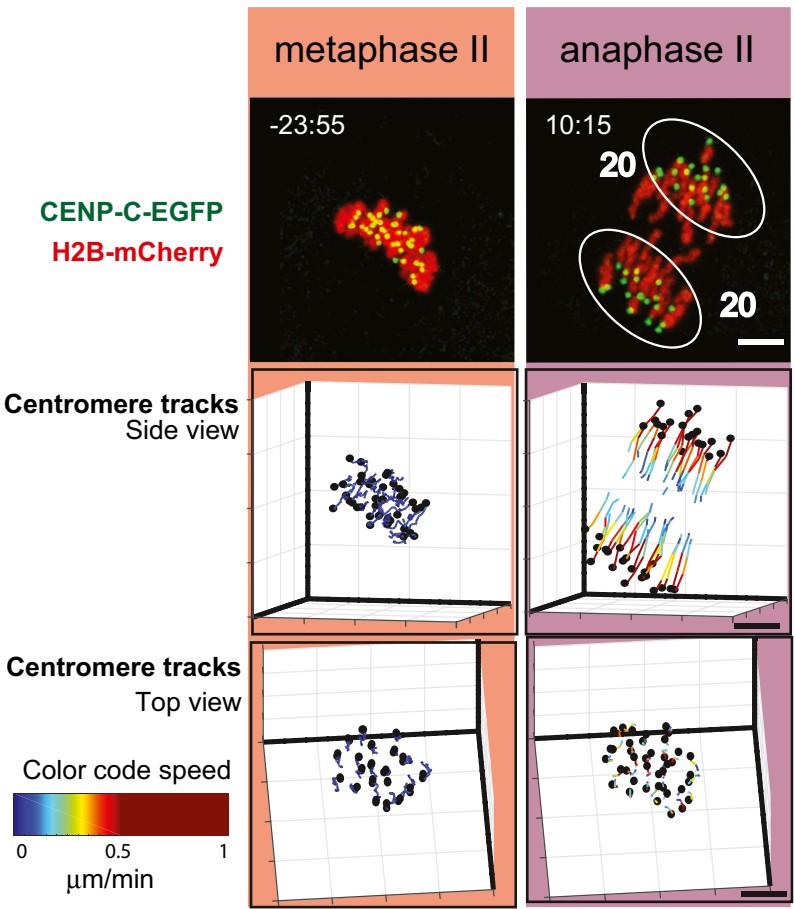

**Figure 3.  3D *in vivo* tracking of chromosome movements from metaphase II to anaphase II.**

Time-lapse imaging of a representative MII oocyte with a normal segregation pattern (i.e. all 20 chromosomes show synchronous equational chromatid separation at anaphase onset, leaving no chromatids behind at the midzone), expressing CENP-C-EGFP (centromeres, green) and H2B-mCherry (chromatin, red). Upper row shows maximum intensity z projection images from representative time points at metaphase II and anaphase II. Number next to the circles enclosing segregating chromatids denotes amount of chromatids segregated to each of the poles. The 3D positions of centromeres are shown as black dots in a side view (along the spindle equator, middle row) and top view (perpendicular to the equator plane, bottom row). Tracks of individual centromeres are colour-coded according to their speed, as indicated by the colour bar. Time is shown relative to anaphase II onset (min:sec). Scale bars, 5 μm.

chromosomes display a reduced level of bi-directional tension, indicative for aberrant merotelic/lateral kinetochore–MT attachments. Though we cannot exclude that the observed dynamic rotation of laggard-producing chromosomes is coupled to attachment error correction, the variability of their inter-centromere distances is similar to what is observed for normally segregating chromosomes (Fig EV5H), indicating a lack of additional attachment correction activity for laggard-producing chromosomes when compared to the normally segregating chromosomes.

We next followed the fate of the 13 laggard-producing chromosomes. We found that only one of the aberrant chromosomes underwent non-disjunction and contributed to formation of an aneuploid gamete (highlighted by darker blue colour in Figs 6 and EV5C–G). The quantitative parameters for the single laggard-producing chromosome that gave rise to non-disjunction at anaphase II stage were not found to be different from the parameters for the other laggard-producing chromosomes.

In summary, more than 95–97.5% of the aberrant kinetochore–MT attachments observed at the metaphase stage are resolved at an early anaphase stage and do not give rise to lagging chromatids in MII oocytes. Furthermore, < 10% (1 out of 13) of laggard-producing chromosomes contribute to formation of aneuploid gametes.

## Discussion

We have here for the first time in a comprehensive manner determined the spatiotemporal segregation pattern for individual chromosomes in mouse MII oocytes. Quantitative datasets of kinetochore dynamics have previously been obtained for human mitotic division (an RPE1 cell line [19], the first meiotic division in mouse female germ cells [18]) and for male meiosis in flies (*Drosophila melanogaster* spermatocytes [23]). Our data, along with the quantitative information obtained from other organisms, are summarized in Table 1.

We find that chromosomes at the beginning of the prometaphase II stage move towards the spindle poles, then return to the spindle equator where they wobble back and forth, perpendicular to the equator plane, before they become stably aligned at the

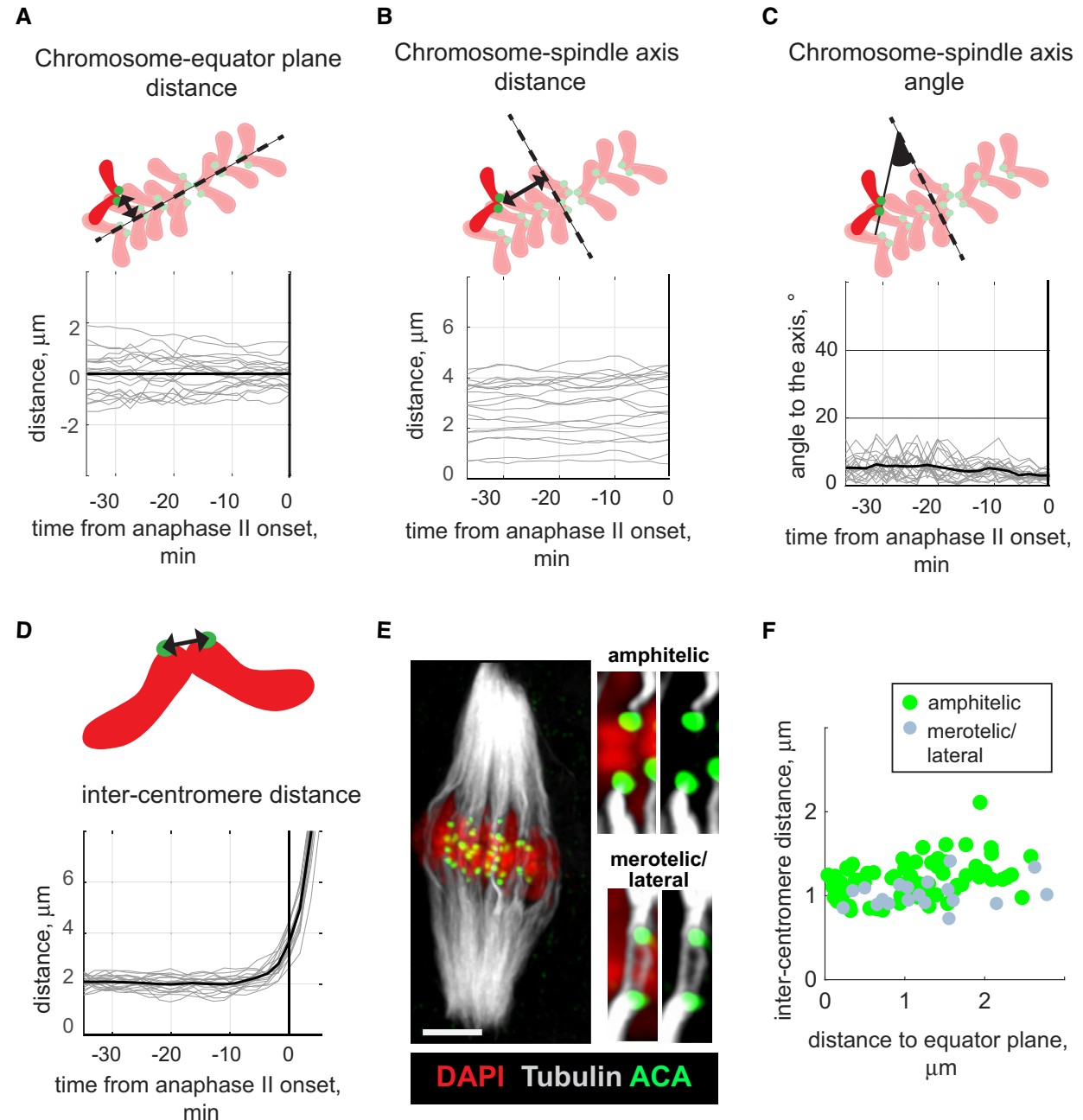

**Figure 4. Chromosomes maintain stable positions and constant inter-centromere distances until anaphase II onset.**

Chromosome parameters obtained for a representative oocyte with normal segregation pattern are shown in (A–D) on the vertical axis, and time in min on the horizontal axis relative to the anaphase II onset. Thin grey lines in (A–D) represent individual chromosomes; thick black line in (C and D) shows the mean value for each time point. On the schemes near the charts, chromosomes are red with green centromeres; spindle equator in (A) and spindle axis in (B and C) are indicated by dashed lines.

A  Changes in distance to the equator plane for all chromosomes in the representative oocyte with normal segregation pattern, shown in Fig 3. The analysed distance is indicated by a black arrowed line on the scheme above the chart.

B  Changes in distance to the spindle axis (shown on the scheme by a black arrowed line) for all chromosomes in oocyte from Fig 3.

C  Changes in chromosome orientation for all chromosomes in oocyte from Fig 3. The orientation is characterized by the angle (highlighted black) between the spindle axis and the line connecting the sister centromeres (black solid line on the scheme).

D  Changes in inter-centromere distance, shown on the scheme by black arrowed line, for all chromosomes in oocyte from Fig 3.

E  Kinetochore–MT attachments in oocytes released from the CSF-dependent metaphase arrest were visualized using an anti-tubulin antibody (white), an anti-centromeric ACA antibody (green) and DAPI to label chromatin (red). The image represents maximum intensity projection through all z-planes containing MTs. Single z-planes of representative chromosomes displaying amphitelic or merotelic/lateral attachments are shown to the right. Scale bar, 10 μm.

F  Chromosomes with amphitelic (green) and merotelic/lateral (grey) attachments were plotted according to their inter-centromere distance (on the horizontal axis) and distance to spindle equator (vertical axis). The inter-centromere distance in fixed oocytes is 1.2 ± 0.2 μm for amphitelic and 1.0 ± 0.2 μm for merotelic attachments (mean ± SD). Data shown for 95 chromosomes taken from 5 oocytes released from the CSF-dependent metaphase II arrest.

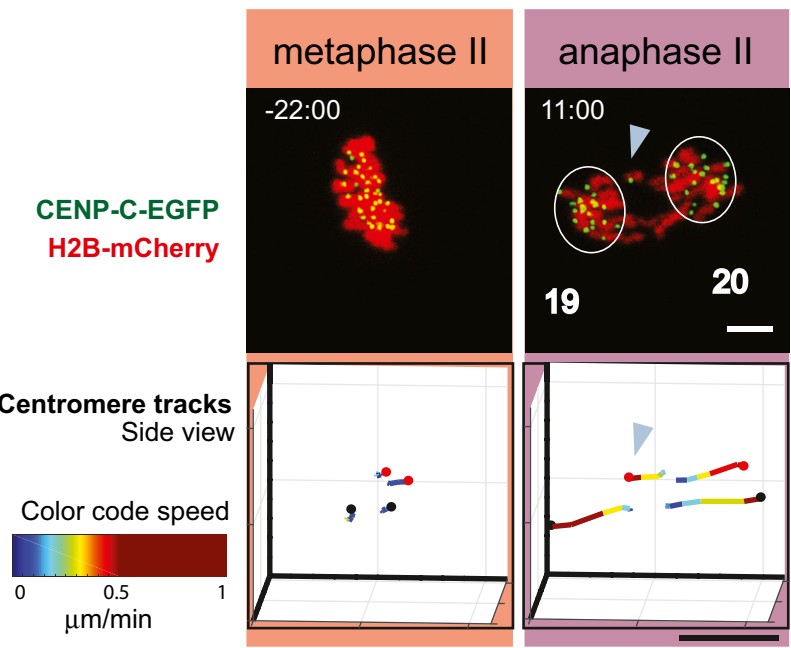

**Figure 5. 3D *in vivo* tracking of chromosome movements in oocytes displaying lagging chromatids at anaphase II.**

A time-lapse imaging of a representative MII oocyte with visible lagging chromatid at midzone during anaphase II expressing CENP-C-EGFP (centromeres, green) and H2B-mCherry (chromatin, red). Upper row shows maximum intensity z projection images from representative time points at metaphase II and anaphase II. A lagging chromatid is indicated by a blue arrowhead. Number next to the circles enclosing the segregating chromatids denotes number of chromatids at each pole. The bottom row depicts 3D positions for centromeres of normally segregating chromosomes as black dots, and centromeres of laggard-producing chromosome as red dots in a side view along the spindle equator. One of the centromeres is left behind at anaphase (blue arrowhead), but will segregate equationally. Tracks of individual centromeres are colour-coded according to the speed, as indicated by the colour bar. Time is shown relative to anaphase II onset (min:sec). Scale bars, 5 μm.

metaphase stage. The inter-centromere distance increases during progression of prometaphase II, from $0.5 \pm 0.1$ μm to $1.6 \pm 0.2$ μm, at the same time as the chromosomes become oriented in parallel with the spindle axis. Thus, initial erroneous syntelic and merotelic/lateral attachments are gradually replaced by the stable end-on amphitelic attachments resulting in the alignment of chromosomes at the metaphase II plate. Remarkably, we observe that more than 20% of chromosomes at the metaphase II stage show merotelic or lateral attachments. The amount of non-amphitelic attachments that we report is in striking contrast to the situation in human somatic cells where about 1% chromosomes show aberrant attachments at the metaphase stage [24]. The lack of a ring-like prepositioning of chromosomes surrounding centrally localized MTs at the prometaphase stage in MII oocytes could contribute to the presence of multiple aberrant kinetochore–MT attachments at the metaphase stage.

The stable positions and constant inter-centromere distances for chromosomes at the metaphase II stage show that kinetochore–MT attachments established by the end of prometaphase II provide constant bi-directional tension on sister kinetochores until anaphase II onset. Importantly, stable positioning and constant inter-centromere distances are also observed for chromosomes with aberrant kinetochore–MT attachments. It is possible that Aurora kinases located at the inter-centromere region of prometaphase and metaphase chromosomes in MII oocytes fail to correct aberrant attachments due to the increased distance observed between sister centromeres in MII chromosomes.

Strikingly, we find that < 1% of the chromosomes in MII oocytes give rise to lagging chromatids at the anaphase stage. It means that very few of the chromosomes with aberrant kinetochore–MT attachments observed at the metaphase stage II contribute to laggard formation. This is comparable to the situation in mitotic cells where 90% of merotelic attachments present at a late metaphase stage do not contribute to laggard formation, possibly a consequence of an unequal number of opposing MTs bound to the single kinetochore of an aberrant chromosome [24]. We also show that < 10% of the lagging chromatids (1 out of 13) that remain at anaphase II undergo non-disjunction and give rise to aneuploid gametes, a result that could be explained by a minor contribution of the MTs that contact the kinetochore from the incorrect side as proposed for the mitotic cells [25,26]. In summary, aberrant kinetochore–MT attachments that accumulate at the metaphase stage are eliminated during the anaphase stage as efficiently in MII oocytes as in mitotic cells.

Interestingly, we find here that laggard-producing chromosomes retain a constant inter-centromere distance and a stable position until anaphase II onset, but fail to keep a parallel orientation to the spindle axis. Furthermore, the inter-centromere distance negatively correlates with the chromosome-axis angle, indicating that reduced inter-centromere tension unexpectedly results in chromosome misorientation. In a recent study, it was proposed that the outer kinetochore of chromosomes during mitosis in human HeLa cells swivels around a CENP-A-containing centromere, facilitating MT capture [27]. Swivelling of the outer kinetochore relative to the

**A**

### Chromosome-equator plane distance for laggard-producing chromosomes

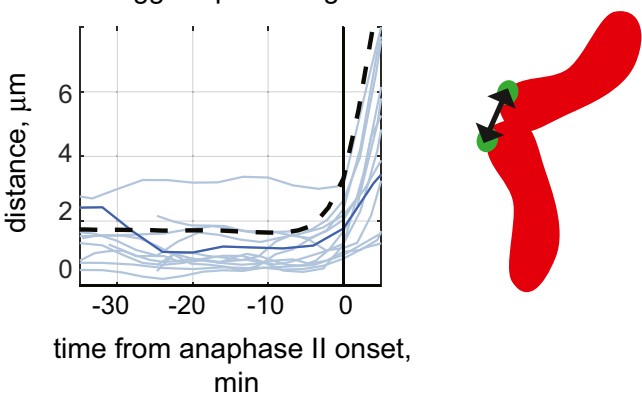

time from anaphase II onset, min

**B**

### Chromosome-spindle axis distance for laggard-producing chromosomes

time from anaphase II onset, min

**C**

### Chromosome-spindle axis angle for laggard-producing chromosomes

time from anaphase II onset, min

**D**

### inter-centromere distance for laggard-producing chromosomes

time from anaphase II onset, min

**Figure 6. Laggard-producing chromosomes at the second meiotic division have an elevated chromosome-axis angle and a reduced inter-centromere distance.**

Chromosome parameters obtained for laggard-producing chromosomes are shown in (A–D) on the vertical axis, and time in min on the horizontal axis relative to the anaphase II onset. Light blue lines indicate 12 laggard-producing chromosomes with correct (equational) chromatid segregation; darker line highlights one chromosome with chromatid non-disjunction. On the schemes near the charts, chromosomes are red with green centromeres; spindle equator in (A) and spindle axis in (B and C) are indicated by black dashed lines.

A Changes in the distance to the equator plane (shown by a black arrowed line on the scheme) for 13 laggard-producing chromosomes. All laggard-producing chromosomes are positioned above the equator at the start of analysis period. Black dotted lines indicate the average thickness of the metaphase plate.
B Changes in the distance to spindle axis (shown by a black arrowed line on the scheme) for 13 laggard-producing chromosomes. Black dotted line indicates the average radius of the metaphase plate.
C Changes in the angle to the spindle axis (highlighted in black on the scheme) for 13 laggard-producing chromosomes. Dashed line on the chart indicates the mean angle for normally segregating chromosomes.
D Changes in the inter-centromere distances (shown by black arrowed line on the scheme) for 13 laggard-producing chromosomes. Dashed line indicates the mean inter-centromere distance for normally segregating chromosomes.

centromere in response to the reduction of tension introduces a rotational moment that could increase the chromosome-axis angle, as we observe here for laggard-producing chromosomes in MII

oocytes. Thus, our data for MII oocytes support and extend the observations previously made for mitotic cells, indicating that chromosomes have a rotational motility. This capacity challenges the

**Table 1. Chromosomes on the metaphase plate in mitosis and meiosis.**

| | Mitosis (in human somatic cells)[a] | Meiosis I (in fly spermatocytes)[b] | Meiosis II (in fly spermatocytes)[b] | Meiosis I (in mouse oocytes)[c] | Meiosis II (in mouse oocytes) | |
| | | | | | Normal chromosomes | Laggard-producing chromosomes |
|---|---|---|---|---|---|---|
| Congression time | ~15 min | ~15 min | ~15 min | ~4 h | 70 ± 20 min | |
| Metaphase time | ~5–10 min | ~15 min | ~15 min | ~4 h | 70 ± 25 min[d] | |
| Max distance from centromere to the equator plane | ~1.5 μm | nd | nd | ~3.5 μm | 1.9 ± 0.6 μm | 2.9 μm |
| Max distance from centromere to the spindle axis | ~5 μm | nd | nd | ~6 μm | 4.6 ± 0.6 μm | 4.5 μm |
| Chromosome-spindle axis angle | ~12° | ~5° | ~5° | ~5° | 6 ± 2° | 17 ± 10° |
| Mean chromosome speed[e] | nd[f] | ~1 μm/min | ~1 μm/min | 0.19 ± 0.05 μm/min | 0.1 ± 0.03 μm/min | 0.1 ± 0.04 μm/min |
| Inter-centromere distance | 0.96 ± 0.21 μm | nd[g] | 0.95 ± 0.1 μm | nd[g] | 1.7 ± 0.2 μm | 1.3 ± 0.7 μm |

[a]In RPE1 cells (data from Magidson et al [19]).
[b]In Drosophila melanogaster spermatocytes (data from Chaurasia & Lehner [23]).
[c]In oocytes derived from 8-week-old mice (data from Kitajima et al [18]).
[d]After release of CSF-dependent arrest.
[e]Averaged for 1.5–5 min.
[f]Nd, not defined.
[g]Sister centromeres are fused together.

model where the inter-centromeric domain is described as an elastic spring [28,29].

The stable nature of chromosome attachments established at the prometaphase stage and maintained until anaphase onset is a unique feature for MII oocytes. It can be speculated that the stability of the formed kinetochore–MT attachments is required to ensure proper chromosome alignment during the extended CSF-mediated metaphase arrest period prior to fertilization. Aberrant kinetochore–MT attachments are not eliminated at the lengthy CSF-mediated metaphase arrest period; instead, more than 99% of the aberrant attachments are resolved correctly during anaphase II, dramatically reducing aneuploidy rate in mouse oocytes. Whether the post-metaphase error correction process reported here is an active mechanism remains an open question for future studies. Impairment of this efficient process could drastically increase aneuploid rate, a situation that could contribute to age-dependent aneuploidy in human oocytes.

# Materials and Methods

## Mouse oocyte culture and microinjection

The animal experiments were approved by the Stockholm-North Animal Ethical Committee and Institutional Animal Care and Use Committee at RIKEN Kobe Branch. Oocytes were taken from 10- to 13-week-old wild-type female mice, produced on a mixed C57BL/6NCrl-129/OlaIHsd background. For immunofluorescence experiments, oocytes were isolated at the germinal vesicle stage and cultured in M2 medium at 37°C; to release the CSF-dependent arrest, oocytes were transferred to G-PGD media (Vitrolife) and

artificially activated by addition of 10 mM SrCl₂ at 37°C. To study the transition period between anaphase I onset and metaphase II arrest by time-lapse imaging, in vitro transcribed 2mEGFP-CENP-C or EGFP-CENP-C together with H2B-mCherry mRNA [18,30] was microinjected at the germinal vesicle stage, and oocytes were matured and imaged in M2 medium at 37°C as described in [Ref. 18,30]. To study the second meiotic division by time-lapse imaging, a reporter gene coding for histone H2B fused to mCherry was introduced into the experimental mouse strain by backcrossing with reporter mice carrying H2B-mCherry fusion gene [31]. In vitro transcribed 2mEGFP-CENP-C mRNA or EGFP-CENP-C was microinjected into CSF-arrested oocytes expressing H2B-mCherry fusion protein. After 2-h incubation in KSOM at 37°C, 5% CO₂ oocytes were activated and imaged in G-PGD media (Vitrolife) supplemented with 10 mM SrCl₂ at 37°C.

## Time-lapse imaging and stage definition

Time-lapse imaging of oocytes was performed using a Zeiss LSM 780 confocal microscope equipped with a 40× C-Apochromat 1.2NA water immersion objective (Carl Zeiss) using the 3D multi-tracking macro [32].We imaged 17–19 consecutive z-confocal sections (512 × 512 pixels, spaced 1.0 or 1.5 μm), at a time interval of 5 min for the period from the anaphase I to CSF-dependent metaphase II arrest period, and at time intervals of 1.5–3 min after release of CSF-dependent arrest. The temporal resolution allowed us to image 5–8 oocytes in the same experiment without apparent phototoxicity effects but also set a limitation for observing movements that lasts for < 5 min in prometaphase II and 1.5–3 min at metaphase II and anaphase II stages. Anaphase onset was set to a

time point when the inter-centromere distance between fused sister chromatids from homologous chromosomes (for anaphase I) or sister centromeres (for anaphase II) started to increase. The period of MI completion comprised both anaphase I and telophase I. The onset and exit from interkinesis was set to a time interval that showed a diffuse H2B-mCherry signal (representing decondensed chromatin). The start of the metaphase II stage was set to a time point when all chromosomes had reached a stable alignment on the metaphase plate.

**Centromere tracking and statistical analysis**

The centromere tracking was performed with Imaris 5.7 image analysis software (Bitplane) using a modified tracking procedure described in [Ref. 18]. The cubic splines with 0.35 smoothing parameter were fitted to centromere tracks, and smoothed values were used for the calculations. To find the position of the centre, we calculated the centroid of all centromeres. At the late prometaphase stage, the direction of the spindle axis was calculated as an average orientation of the chromosomes that showed more than 70% of their inter-centromere distance at the metaphase stage. At the early prometaphase stage with < 10 chromosomes showing > 70% of their metaphase inter-centromere distance, we used an averaged axis orientation calculated for the first 3 time points when the axis orientation could be defined as described above as an estimation of the axis orientation at the earlier time points. At the metaphase stage, the spindle axis was defined as a line that had an averaged orientation of all aligned chromosomes. The spindle axis went through the centre, and the spindle equator plane was perpendicular to the spindle axis and crossed it at the centre. The chromosome position was defined as a midpoint of the line connecting two sister centromeres. The chromosome orientation was defined as the angle between the spindle axis and the line that connected sister centromeres. The data processing and plotting was performed with the help of Fiji [33], MATLAB (Bitplane) and GraphPad Prism (GraphPad Software, Inc.). The statistical analysis was performed by GraphPad Prism and R software (https://www.r-project.org). For statistical analysis, we used Pearson's correlation test to probe correlation between chromosome orientation and inter-centromere distance in laggard-producing chromosomes; two-way ANOVA to compare the chromosome angle, chromosome speed and inter-centromere distance of laggard-producing chromosomes with normally segregating chromosomes; and nested ANOVA to compare the chromosome parameters between oocytes.

**Oocyte fixation and immunofluorescent imaging**

Microtubules were stabilized by fixation in 1.9% formaldehyde in BRD80 buffer (80 mM K-PIPES, 1 mM $MgCl_2$, 1 mM EGTA, pH 6.8) after 5 min cold treatment in 80 mM K-PIPES, 1 mM $MgCl_2$, pH 7.4, as described in [Ref. 16]. Prometaphase II oocytes were fixed 1.5–2.5 h after the first polar body extrusion. To obtain chromosomes at the CSF-dependent metaphase arrest stage, we fixed oocytes 6-8 h after the first polar body extrusion. Chromosomes at the metaphase II stage after release from the CSF-dependent arrest were fixed 40 min after activation. The Aurora B and C kinases were probed after spreading in 1% PFA, as described in [Ref. 34]. The centromeres were labelled by ACA (Antibodies Inc.) 1:100, microtubules were visualized by Tubulin-FITC (Sigma) 1:2,000, Aurora B was probed by rabbit anti-Aurora B antibody (own production against peptide GLNTLSQRVLRKEPATTSALA) at 1:50 dilution and Aurora C by guinea pig anti-Aurora C antibody (own production against peptide PGGELYKELQRHQKLDQQRT) at 1:50 dilution; the secondary antibodies were swine-anti-rabbit FITC (DakoCytomation) at 1:400 dilution, donkey-anti-guinea pig Alexa 546 (Invitrogen) at 1:1,000 dilution and donkey-anti-human Alexa 647 (Invitrogen) at 1:100 dilution. Oocytes were mounted in ProLong Gold (Thermo Fisher Scientific). Images with kinetochore–MT attachments were collected at the Zeiss 800 with an Airyscan module at 63×/1.4 NA objective, and the Aurora B and Aurora C were visualized after collection at Leica DMRX at 100× 1.4/NA objective. Images were processed by ZEN blue software with Airyscan processing module, Volocity (Improvision) and Imaris 5.7 (Bitplane).

**Expanded View** for this article is available online.

## Acknowledgements

A.K. and C.H. are supported by grants from Horizon 2020 (634113), Vetenskapsrådet (Swedish Research Council; K2013-54X-21397-04-5), Cancerfonden (Swedish Cancer Society; 170226) and the Karolinska Institutet. T.S.K is supported by grant from JSPS KAKENHI 24770173/16H06161, and H.B. is supported by grant from National Microscopy Infrastructure, NMI (VR-RFI 2016-00968). We thank J. G. Liu for help with breeding mice and mRNA microinjections and S. Valentiniene for technical support. We thank G. Månsson and C. Edwards from the CLICK imaging facility at Karolinska Institutet for assistance with the Imaris software.

## Author contributions

AK designed and conducted the experiments, analysed the data and wrote the manuscript; TSK developed the analysis procedure and conducted some of the experiments; HB provided expert assistance related to time-lapse microscopy; CH designed the experiments and wrote the manuscript.

## Conflict of interest

The authors declare that they have no conflict of interest.

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
