## [Review Process File · EMBO Reports]

Post-metaphase correction of aberrant kinetochore-microtubule attachments in mammalian eggs

Anna Kouznetsova, Tomoya S. Kitajima, Hjalmar Brismar, Christer Höög

Review timeline:

Submission date:	11 February 2019
Editorial Decision:	6 March 2019
Revision received:	3 April 2019
Editorial Decision:	3 May 2019
Revision received:	24 May 2019
Accepted:	7 June 2019

Editor: Deniz Senyilmaz Tiebe

Transaction Report:

1st Editorial Decision6 March 2019

Thank you for submitting your manuscript for consideration by EMBO Reports. It has now been seen by two referees whose comments are shown below.

As you can see, both referees express interest in your study describing the movements of chromosomes during meiosis in mouse oocytes. However, they also raise concerns that need to be addressed in full before we can consider publication of the manuscript here.

Given these constructive comments, I would like to invite you to revise your manuscript with the understanding that the referee must be fully addressed and their suggestions taken on board. Please address all referee concerns in a complete point-by-point response. Acceptance of the manuscript will depend on a positive outcome of a second round of review. It is EMBO Reports policy to allow a single round of revision only and acceptance or rejection of the manuscript will therefore depend on the completeness of your responses included in the next, final version of the manuscript.

Supplementary/additional data: The Expanded View format, which will be displayed in the main HTML of the paper in a collapsible format, has replaced the Supplementary information. You can submit up to 5 images as Expanded View. Please follow the nomenclature Figure EV1, Figure EV2 etc. The figure legend for these should be included in the main manuscript document file in a section called Expanded View Figure Legends after the main Figure Legends section. Additional

Supplementary material should be supplied as a single pdf labeled Appendix. The Appendix includes a table of content on the first page with page numbers, all figures and their legends. Please follow the nomenclature Appendix Figure Sx throughout the text and also label the figures according to this nomenclature. For more details please refer to our guide to authors.

When preparing your letter of response to the referees' comments, please bear in mind that this will form part of the Review Process File, and will therefore be available online to the community. For more details on our Transparent Editorial Process, please visit our website:
http://emboj.embopress.org/about#Transparent_Process

Regarding data quantification, please ensure to specify the name of the statistical test used to generate error bars and P values, the number (n) of independent experiments underlying each data point (not replicate measures of one sample), and the test used to calculate p-values in each figure legend. Discussion of statistical methodology can be reported in the materials and methods section, but figure legends should contain a basic description of n, P and the test applied. Please also include scale bars in all microscopy images.

We now strongly encourage the publication of original source data with the aim of making primary data more accessible and transparent to the reader. The source data will be published in a separate source data file online along with the accepted manuscript and will be linked to the relevant figure. If you would like to use this opportunity, please submit the source data (for example scans of entire gels or blots, data points of graphs in an excel sheet, additional images, etc.) of your key experiments together with the revised manuscript. Please include size markers for scans of entire gels, label the scans with figure and panel number, and send one PDF file per figure.

- a complete author checklist, which you can download from our author guidelines (<http://embor.embopress.org/authorguide#revision>). Please insert page numbers in the checklist to indicate where the requested information can be found.
 - a letter detailing your responses to the referee comments in Word format (.doc)
 - a Microsoft Word file (.doc) of the revised manuscript text
 - editable TIFF or EPS-formatted figure files in high resolution
- (In order to avoid delays later in the publication process please check our figure guidelines before preparing the figures for your manuscript:
http://www.embopress.org/sites/default/files/EMBOPress_Figure_Guidelines_061115.pdf)
- a separate PDF file of any Supplementary information (in its final format)
 - all corresponding authors are required to provide an ORCID ID for their name. Please find instructions on how to link your ORCID ID to your account in our manuscript tracking system in our Author guidelines (<http://embor.embopress.org/authorguide>).

As part of the EMBO publication's Transparent Editorial Process, EMBO reports publishes online a Review Process File to accompany accepted manuscripts. This File will be published in conjunction with your paper and will include the referee reports, your point-by-point response and all pertinent correspondence relating to the manuscript.

I look forward to seeing a revised version of your manuscript when it is ready. Please let me know if you have questions or comments regarding the revision.

REFeree REPORTS

Referee #1:

The authors describe in a very detailed manner with unprecedented resolution chromosome movements of mouse oocytes progressing from meiosis I into meiosis II, remaining arrested in meiosis II, and after release from arrest. They show that a high fraction of chromosomes are not correctly aligned in metaphase II, being attached in a merotelic or lateral manner. Surprisingly though, this does not lead to the same elevated missegregation rate in anaphase II. This study will move the field forward in a significant manner, and its descriptive nature should not be an argument for rejection.

Minor points:

1) It is not clear whether the 20 % of chromosomes that are found not the aligned correctly are detected in all oocytes, or only in a fraction of oocytes.

2) The authors suggest that there is a mechanism correcting attachments at anaphase II onset. Maybe attachments are merotelic or lateral, but this is "just good enough" for correct segregation in most cases? I think it is premature to propose an active mechanism for correction in anaphase II.

Referee #2:

Kouznetsova et al. describe the behavior of kinetochores (KTs)/chromosomes during completion of the meiotic divisions in normal mouse oocytes, including time lapse imaging as developed and used earlier for a study on meiosis I (Kitajima et al. 2011). Here the focus is on the period from anaphase I to anaphase II, which includes the arrest in metaphase II that occurs in mammalian oocytes until fertilization. The time lapse imaging provides insight into the temporal and spatial dynamics governing the integration of KT/chromosomes into meiosis II spindles as well their segregation during anaphase II. Time lapse imaging is complemented with an analysis of oocytes that were fixed and stained at time points of interest after brief incubation at 4°C for elimination of those microtubules (MTs) that are not cold-stable. MTs attached to KT are well-known to be cold-stable, but some other MTs (primarily interpolar) survive the cold-treatment as well, as also evident from this work. Based on the analysis of fixed oocytes, Kouznetsova et al. classify KT-MT interactions and chromosome integration into the meiosis II spindle as either amphitelic, syntelic or merotelic/lateral.

To my knowledge the time lapse analysis described here is unprecedented with regard to detail. Kouznetsova et al. (2014) have already applied time lapse imaging of chromosomes (marked by H2B-mCherry). But here KT (marked by Cenp-C-eGFP) are imaged as well, followed by KT tracking. The resulting descriptive data on durations of exit from MI, interkinesis, prometaphase II, and anaphase II (after experimentally breaking the metaphase II arrest by addition of SrCl₂), as well as on the speed of chromosome movements and the separation distance between sister KT during different phases will certainly be of high value for researchers in the field of mouse oocyte meiosis. As described below, some higher temporal resolution might be required to exclude some potential underestimates. Of similar value is the convincingly documented observation that chromosomes are not arranged into a transient equatorial belt during meiosis II, in contrast to meiosis I and mitosis where this arrangement has been proposed to increase speed and accuracy of the chromosome bi-orientation process. Moreover, the demonstration that KT/chromosomes remain stably attached to the spindle throughout the metaphase II arrest is also revealing, as it contrast with the relatively frequent re-orientations that were observed throughout prometaphase I and metaphase I. The manuscript of Kouznetsova et al. puts the main emphasis on the apparent striking difference in the frequencies of erroneous KT attachments during metaphase II and of lagging chromosomes during anaphase II. However, as detailed below, this might represent at least in part an unjustified overemphasis based on scoring of KT/chromosome attachment types with a procedure that has limited accuracy. Moreover, if the findings of Kouznetsova et al. were indeed accurate, the main emphasis should be given to the apparent absence of error correction throughout the metaphase arrest (in contrast to the notion that error correction is active during mitotic metaphase) rather than on post-metaphase correction of the errors. In case of mitosis, it has been well established since many years that merotelic attachments are frequently corrected post-metaphase and that therefore they do not necessarily lead to much lagging and missegregation (Cimini et al. 2003, abstract: "... Surprisingly, anaphase lagging chromosomes represented a very small fraction of merotelic kinetochore orientations present in late metaphase. ...").

Overall this study is largely convincing technically. It provides very useful descriptive information. The presentation seems to have a certain tendency to over-exaggerate some aspects that might actually be less unusual than claimed.

Specific comments:

A. Major comments

1. An almost identical procedure for scoring KT attachments during the metaphase II arrest has been used earlier by Zhang et al. 2017. JCB. These authors report that about 95% of the chromosomes have the correct amphitelic attachment in the metaphase II arrest. Here the fraction is reported to be about 5 times higher. The discrepancy likely reflects a problem with the assay: 1. cold treatment does not neatly remove all MTs except those attached to the KT. 2. Light microscopy provides insufficient resolution to distinguish MTs as either true KT-MTs (i.e., bound by KT proteins) or apparent KT-MTs (i.e., bundled with other MTs including KT-MTs and passing close to the KT but nevertheless too far for direct association with KT proteins.). The reasonable correlation between type of attachment and separation distance between sister KTs (here see Fig. 2E and 4F) argues that overall this popular scoring approach is not without merit. Overall, those scored as "syntelic" and "merotelic/lateral" are associated with lesser sister KT separation than the "amphitelic". Nevertheless, there is considerable overlap between "amphitelic" and in particular "merotelic/lateral". On the one hand, this might reflect that only few of the MTs generate the merotelic while the great majority connecting to a sister KT pair is attached correctly (amphitelic). The fact that such chromosomes will not display lagging and/or missegregation is hardly surprising. On the other hand, erroneous scoring might contribute. Unfortunately, a simple more reliable procedure for better scoring of the type of KT/chromosome attachments is not available to my knowledge. CLEM would be more reliable but highly demanding. At a minimum, therefore, Kouznetsova et al. should mention and discuss the apparent discrepancy with Zhang et al. (2017).

2. p.14: "We also show that less than 10% of the lagging chromatids that remain at anaphase II undergo nondisjunction and give rise to aneuploid gametes,"
The statistical basis for this conclusion is insufficient. 1 of 13 observed laggards was missegregated. If 2 or 3 of the observed laggards would have given rise to missegregation, there would have been more than 10% (15 or 23%, respectively). I am not convinced that an analysis of the next 13 laggards would again detect just one missegregation event. Perhaps more, perhaps less. Kouznetsova need to analyze far more oocytes or qualify their conclusion adequately.

3. Fig. 6C presents curves representing the angle between the spindle axis and the inter-sister KT axis of laggard producing chromosomes. Accordingly, at least some of the lagging chromosomes appear to be subject to a correction process before anaphase II onset (angle changing from a large to a normal small value). Does this indicate that error correction is more active already before anaphase II than what the manuscript appears to suggest?

4. The reported speed of KTs during prometaphase II is about one order of magnitude below that observed in insects during the corresponding phase (for example Church and Lin, 1985; Savoian et al., 2000; Chaurasia and Lehner, 2018). Perhaps this striking difference is lower than apparent. The interval between the z-stack acquisition time points chosen by Kouznetsova et al. is 5 minutes and hence far longer than that used in other organisms. As a result, rapid movements on lower spatial scales might be missed. Additional analysis of prometaphase II with considerably higher temporal resolution should be included. Moreover, it should be very helpful for increasing general interest when a comparative discussion of the findings made in mouse oocytes with those in other organisms was included.

Kouznetsova include comparisons between meiosis II and mitosis. Unfortunately, they compare data from different species: data from mouse in case of meiosis II with data from human RPE1 cells Magidson et al. 2011). This might be misleading (centromere DNA sequences and KT proteins evolve rapidly). Presumably, adequate data concerning mouse mitosis has not yet been published, but it should not be very difficult to generate.

B. Minor comments:

5. Scoring the onset of interkinesis accurately seems quite impossible when looking at Suppl Movie 1. How was this actually done? How reliable is the corresponding data.

6. p. 10: "Inter-centromere distances for aberrantly attached chromosomes were reduced in comparison to chromosomes with amphitelic attachments (Fig.4F, $p=0.001$, Nested ANOVA) validating that merotelic/lateral attachments contribute to a reduced level of bi-directional tension." Please provide the average separation of observed in those with normal amphitelic attachment and those with mero/lateral. The difference between the two types appears less than in metaphase II arrest, perhaps hinting at some error correction in the metaphase II arrest.

7. Vallot et al. 2018 and Kouzmetsova et al. 2014 are not included in reference list

8. Typos: microscope, Pierson's,

9. Introduction, sentence 1: not entirely accurate. There are eukaryotes like budding yeast which progress through mitosis in the haploid states; so mitosis can generate haploid daughter cells.

10. Introduction: "..., whereas merotelic attachments are often seen during meiosis I (MI) (Yoshida et al, 2015; Zielinska et al, 2015) and meiosis II (MII) (Zhang et al, 2017)."

As already indicated above, Zhang et al. 2017 detect some merotelic attachment in control oocytes but clearly considerably less than what is reported in this study. Citing this reference as showing frequent merotelic attachment seems inappropriate.

1st Revision - authors' response

3 April 2019

Referee #1:

Minor points:

1) It is not clear whether the 20 % of chromosomes that are found not the aligned correctly are detected in all oocytes, or only in a fraction of oocytes.

20% of chromosomes with erroneous (non-amphitelic) attachments are found in each oocyte.

We changed the concluding sentence on page 10 to make this point clear.

page 10: "... As a result of this, erroneous attachments formed at the prometaphase II stage, affecting more than 20% of the chromosomes in each oocyte at the MII stage are not corrected prior to anaphase onset."

2) The authors suggest that there is a mechanism correcting attachments at anaphase II onset. Maybe attachments are merotelic or lateral, but this is "just good enough" for correct segregation in most cases? I think it is premature to propose an active mechanism for correction in anaphase II.

We agree with the reviewer that our results do not provide information on the details of how correction is achieved. We therefore refer to a correction process, rather than correction mechanism, in the manuscript. We have added a sentence to the final paragraph on page 16 to make this point clear.

page 16: "... Whether the post-metaphase error correction process reported here is an active mechanism remains an open question for future studies. Impairment of this efficient process could drastically increase aneuploid rate, a situation that could contribute to age-dependent aneuploidy in human oocytes."

Referee #2:

A. Major comments

1. An almost identical procedure for scoring KT attachments during the metaphase II arrest has been used earlier by Zhang et al. 2017. JCB. These authors report that about 95% of the chromosomes have the correct amphitelic attachment in the metaphase II arrest. Here the fraction is reported to be about 5 times higher. The discrepancy likely reflects a problem with the assay: 1. cold treatment does not neatly remove all MTs except those attached to the KT. 2. Light microscopy provides

insufficient resolution to distinguish MTs as either true KT-MTs (i.e., bound by KT proteins) or apparent KT-MTs (i.e., bundled with other MTs including KT-MTs and passing close to the KT but nevertheless too far for direct association with KT proteins.). The reasonable correlation between type of attachment and separation distance between sister KTs (here see Fig. 2E and 4F) argues that overall this popular scoring approach is not without merit. Overall, those scored as "syntelic" and "merotelic/lateral" are associated with lesser sister KT separation than the "amphitelic". Nevertheless, there is considerable overlap between "amphitelic" and in particular "merotelic/lateral". On the one hand, this might reflect that only few of the MTs generate the merotelic while the great majority connecting to a sister KT pair is attached correctly (amphitelic). The fact that such chromosomes will not display lagging and/or missegregation is hardly surprising. On the other hand, erroneous scoring might contribute. Unfortunately, a simple more reliable procedure for better scoring of the type of KT/chromosome attachments is not available to my knowledge. CLEM would be more reliable but highly demanding. At a minimum, therefore, Kouznetsova et al. should mention and discuss the apparent discrepancy with Zhang et al. (2017). **The referee is correct that the difference in the protocols could explain the discrepancy between our results (~20% of attachments are merotelic/lateral) and the results reported in Zhang et al., 2017 (~5% of all attachments in mature metaphase II oocytes are merotelic/lateral). We have tested the fixation protocol described in Zhang et al., 2017. We found however, that a majority of the kinetochores became detached from the microtubules when we followed this protocol. We instead used the protocol by Vallot et al., 2018.**

To preserve the MTs, we performed the cold treatment in microtubule-stabilizing buffer instead of M2 medium and the fixation was performed in 1.9% formaldehyde in microtubule-stabilizing buffer instead of 4% PFA in PBS. In addition, for image acquisition we used Zeiss LSM800 microscope equipped with the Airyscan module (Huff et al., 2015, Nature Methods). This module has 4-8 times better signal to noise ratio and 1.7x better resolution comparing to the traditional LSM systems with a 1 AU pinhole; resulting in resolution up to 120nm in xy- and 350nm in z-dimensions. Such resolution allows improved visualization of fine MTs and exclusion of non-KT MT. Zhang et al. scored ~84% of the attachments present in mature metaphase II oocytes, while we scored and classified 95-98% of the attachments.

An explanation to the difference between our results and the data published in Zhang et al., 2011 has been added to page 13 of the manuscript:

page 13: "...Remarkably, we observe that more than 20% of chromosomes at the metaphase II stage show merotelic or lateral attachments. This is more than previously reported (about 5% of attachments were scored as merotelic or lateral in (Zhang et al, 2017). The discrepancy is most probably explained by a different visualization protocol, where we employed a stabilizing buffer to preserve MTs and an AiryScan microscope module (Zeiss) to achieve super-resolution. "

2. p.14: "We also show that less than 10% of the lagging chromatids that remain at anaphase II undergo nondisjunction and give rise to aneuploid gametes,"

The statistical basis for this conclusion is insufficient. 1 of 13 observed laggards was missegregated. If 2 or 3 of the observed laggards would have given rise to missegregation, there would have been more than 10% (15 or 23%, respectively). I am not convinced that an analysis of the next 13 laggards would again detect just one missegregation event. Perhaps more, perhaps less. Kouznetsova need to analyze far more oocytes or qualify their conclusion adequately.

We agree with the reviewer that the numbers of lagging and missegregated chromatids analyzed are too low to provide a precise quantitative number. We have changed the text on page 14 to be as accurate as possible.

page 14: "...We also show that less than 10% of the lagging chromatids (1 out of 13) that remain at anaphase II undergo nondisjunction and give rise to aneuploid gametes,...."

3. Fig. 6C presents curves representing the angle between the spindle axis and the inter-sister KT axis of laggard producing chromosomes. Accordingly, at least some of the lagging chromosomes appear to be subject to a correction process before anaphase II onset (angle changing from a large to a normal small value).

We have performed the statistical analysis of the chromosome-axis angle of laggard-producing chromosomes at the frame preceding anaphase onset. This angle is statistically larger than the average value for normally segregating chromosomes (p=0.01, one sample t-test).

Does this indicate that error correction is more active already before anaphase II than what the manuscript appears to suggest?

A conventional indicator of attachment correction at the metaphase II stage is the distance between sister centromeres: it increases when tension is increased (i.e. when merotelic attachments are converted to correct amphitelic attachments) and remains stable in the absence of efficient correction. We observed that this distance does not increase before the anaphase II onset for both normally segregating chromosomes (Fig.4D) and laggard-producing chromosomes with presumably merotelic attachments (Fig.6D). Furthermore, inter-centromere distances observed for normally segregating chromosomes and laggard producing chromosomes shows equal variability (a newly added Fig. EV5H), indicating an absence of additional correction activity for laggard-producing chromosomes. Thus, we used the stability of inter-centromere distance as a rationale for our conclusion on the absence of attachment correction prior to anaphase II onset.

We had added new explanatory text to pages 12 and 14 of the manuscript and a new panel in supplementary figure EV5 demonstrating the equal variability of inter-centromere distances for normally segregating and laggard-producing chromosomes:

page 12: "...Notably, despite dynamic rotation of laggard-producing chromosomes, the variability of their inter-centromere distances is similar to normally segregating chromosomes (Fig. EV5H), indicating the absence of additional attachment correction activity in laggard-producing chromosomes."

page 14: "...Importantly, stable positioning and constant inter-centromere distances are also observed for chromosomes with aberrant kinetochore-MT attachments."

4. The reported speed of KT's during prometaphase II is about one order of magnitude below that observed in insects during the corresponding phase (for example Church and Lin, 1985; Savoian et al., 2000; Chaurasia and Lehner, 2018). Perhaps this striking difference is lower than apparent. The interval between the z-stack acquisition time points chosen by Kouznetsova et al. is 5 minutes and hence far longer than that used in other organisms. As a result, rapid movements on lower spatial scales might be missed. Additional analysis of prometaphase II with considerably higher temporal resolution should be included.

We agree with the reviewer that higher temporal resolution would yield more information about KT dynamics at prometaphase II, as we do not detect any rapid movements that last for less than 5 min in prometaphase II and 1.5-3 min at metaphase II. However, obtaining a higher temporal resolution in mouse oocytes is difficult of several reasons: (i) meiosis in mouse oocytes is prolonged in comparison to other organisms studied by the same method. It takes 10-12 hours to reach the metaphase II arrest stage in mouse oocytes and 70 min to reach anaphase II after resuming meiosis. In comparison, mitosis lasts for ~30 min in human and mouse somatic cells, and both meiotic divisions are completed in ~3 hours in fly spermatocytes. Imaging the prolonged processes in mouse oocytes with significantly higher temporal resolution will inevitably result in artifacts due to phototoxicity; (ii) mouse oocytes have diameter of about 80um. To reduce phototoxicity and increase resolution, we have imaged a 30x30x18um region centered on the chromosomes and used post-acquisition processing to follow chromosome movements inside the oocytes. It takes ~30 sec to collect one time point for one oocyte with the quality required for a reliable tracking of all centromeres at the second meiotic division.

Notably, the KT speed we observe at prometaphase II with 5 min temporal resolution is in excellent agreement with the speed of the KT's observed at prometaphase I by Kitajima, et al., 2011 with 1.5 min resolution.

We have added an explanation for our choice of temporal resolution and the limitations of the approach to the manuscript on page 17:

page 17: "...The temporal resolution allowed us to image 5-8 oocytes in the same experiment without apparent phototoxicity effects but also set a limitation for observing movements that lasts for less than 5 min in prometaphase II and 1.5-3 min at metaphase II and anaphase II stages."

Moreover, it should be very helpful for increasing general interest when a comparative discussion of the findings made in mouse oocytes with those in other organisms was included.

We agree with the reviewer and have also included KT tracking datasets for *Drosophila melanogaster* spermatocytes (Chaurasia & Lehner, 2018). We have included the data in Table 1 and in the Discussion, page 13:

page 13: "...Quantitative datasets of kinetochore dynamics have previously been obtained for human mitotic division (an RPE1 cell line, Magidson et al, 2011), the first meiotic division in mouse female germ cells (Kitajima et al, 2011) and for male meiosis in flies (Drosophila melanogaster spermatocytes, Chaurasia & Lehner, 2018). Our data, along with the quantitative information obtained from other organisms, is summarised in Table 1."

Kouznetsova include comparisons between meiosis II and mitosis. Unfortunately, they compare data from different species: data from mouse in case of meiosis II with data from human RPE1 cells (Magidson et al. 2011). This might be misleading (centromere DNA sequences and KT proteins evolve rapidly). Presumably, adequate data concerning mouse mitosis has not yet been published, but it should not be very difficult to generate.

The reviewer is correct that high-resolution quantitative data on mouse mitosis is not available (or at least we can not find it in the literature). Published data, however, on the duration of mitotic prometaphase and metaphase stages (~15 min and ~5-10 min, e.g. Moon et al, 2014) or inter-kinetochore distance (~0.8 μ m, e.g. Lica et al., 1986) for mouse mitotic cells are available and are quite similar to the results obtained in human cells. We find it therefore reasonable considering the scope of this study to compare our data to the high-resolution imaging data that had been obtained using tracking approach similar to ours (Magidson et al., 2011).

We have included the available quantitative data on mouse mitosis.

page 6: "...but much slower than reported for human and mouse somatic cells (about 15 min, Magidson et al, 2011; Moon et al, 2014).

page 7: "...a chromosome arrangement described at the prometaphase stage in human mitotic cells (present 1.5 to 10 min after NEBD) and at the prometaphase stage in mouse MI oocytes (present 30 to 120 min after GVBD),..."

page 9: "...The metaphase chromosomes were oriented almost parallel to the spindle axis with a mean chromosome-axis angle of $6\pm 2^\circ$ (Fig 4C; Fig EV3 B, C), comparable to the chromosome-axis angle observed at the metaphase stage in MI oocytes (Kitajima et al, 2011), but slightly smaller than observed at the metaphase stage in human mitotic cells (Magidson et al, 2011). The inter-centromere distance for chromosomes at the metaphase II stage was $1.6\pm 0.2 \mu$ m, almost twice as large as what was observed at the metaphase stage in human and mouse mitotic cells (Magidson et al, 2011; Lica et al, 1986) and did not change until anaphase II onset (Fig 4D, Fig EV3 D, E)."

B. Minor comments:

5. Scoring the onset of interkinesis accurately seems quite impossible when looking at Suppl Movie 1. How was this actually done? How reliable is the corresponding data.

We agree with the reviewer that the scoring of interkinesis onset can be performed only with limited resolution in the absence of other markers than H2B-mCherry and CENP-C-EGFP. The onset of interkinesis was scored when the H2B-mCherry signal became diffuse and chromosomes completely lost individualization. Chromatin does not attain a completely round shape during interkinesis, probably because the nuclear envelope does not form in mouse oocytes (Holt, Lane and Jones, 2013). A more detailed description for how stage determination was carried out has now been added to Material and Methods Section (page 17).

page 17: "...Anaphase onset was set to a timepoint when the inter-centromere distance between fused sister chromatids from homologous chromosomes (for anaphase I) or sister centromeres (for anaphase II) started to increase. The period of MI completion comprised both anaphase I and telophase I. The onset and exit from interkinesis was set to a time interval that showed a diffuse H2B-mCherry signal (representing decondensed chromatin). The start of the metaphase II stage was set to a time point when all chromosomes had reached a stable alignment on the metaphase plate. "

6. p. 10: "Inter-centromere distances for aberrantly attached chromosomes were reduced in comparison to chromosomes with amphitelic attachments (Fig.4F, $p=0.001$, Nested ANOVA) validating that merotelic/lateral attachments contribute to a reduced level of bi-directional tension."

Please provide the average separation of observed in those with normal amphitelic attachment and those with mero/lateral. The difference between the two types appears less than in metaphase II arrest, perhaps hinting at some error correction in the metaphase II arrest.

The average inter-centromere distance obtained by immunostaining experiments in fixed oocytes after the release of CSF-dependent metaphase II arrest (Fig.4F) is 1.2 ± 0.2 μm for amphitelic and 1.0 ± 0.2 μm for merotelic/lateral attachments. We have added the values to the figure legend.

It should be noted, that the average values for inter-centromere distances in live oocytes are 1.7 ± 0.2 μm for normally segregating chromosomes and 1.3 ± 0.7 μm for laggard-producing chromosomes with erroneous merotelic attachments (Fig. 6D). Both absolute values and the difference for the two types of attachments increases in comparison to the values obtained for fixed oocytes. The difference in inter-centromere distances measured in fixed and live oocytes is most probably explained by distortions inevitably resulting from the fixation procedure. Importantly, we observe constant inter-centromere distances in live imaging experiments (Fig. 2C, Fig.4D, Fig.6D), which argue against error correction taking place during the CSF-dependent metaphase II arrest or after arrest release.

7. Vallot et al. 2018 and Kouzmetsova et al. 2014 are not included in reference list
We thank the reviewer for pointing this out, the reference list has been corrected.

8. Typos: microscope, Pierson's,
We thank the reviewer for pointing this out, the typos have been corrected.

9. Introduction, sentence 1: not entirely accurate. There are eukaryotes like budding yeast which progress through mitosis in the haploid states; so mitosis can generate haploid daughter cells.
We thank the reviewer for pointing this out. We have changed the word "eukaryotes" to "mammals" to make the statement more accurate.

10. Introduction: "..., whereas merotelic attachments are often seen during meiosis I (MI) (Yoshida et al, 2015; Zielinska et al, 2015) and meiosis II (MII) (Zhang et al, 2017)."
As already indicated above, Zhang et al. 2017 detect some merotelic attachment in control oocytes but clearly considerably less than what is reported in this study. Citing this reference as showing frequent merotelic attachment seems inappropriate.
We thank the reviewer for pointing this out. We have removed the word "often" to make the statement more accurate.

2nd Editorial Decision

3 May 2019

Thank you for submitting your revised manuscript. It has now been seen by both of the original referees. As you can see, referees find that their concerns have been sufficiently addressed and in principle recommend publication. However, before I can send the acceptance letter, there are some editorial concerns I need you to address.

- Please address the remaining concerns of referee #2 by performing the recommended textual changes. Please let me know if you would like to discuss this further.
- Please provide 3-5 keywords for your study. These will be visible in the html version of the paper and on PubMed and will help increase the discoverability of your work.
- Our production/data editors have asked you to clarify several points in the figure legends (see attached document). Please incorporate these changes in the attached word document and return it with track changes activated.
- Please provide a brief running title for your manuscript.
- We noticed that the movies are currently not called out in the text.
- We realized that the reference format is currently incorrect. Please refer to <http://embor.embopress.org/authorguide#referencesformat>
- Movie legends should be removed from the main text. Each movie legend should be zipped with

the respective movie and uploaded as individual zip files.

- Papers published in EMBO Reports include a 'Synopsis' to further enhance discoverability. Synopses are displayed on the html version of the paper and are freely accessible to all readers. The synopsis includes a short standfirst as well as 2-5 one sentence bullet points that summarise the paper and are provided by the authors. I would therefore ask you to include your suggestions for bullet points.
- In addition, please provide an image for the synopsis. This image should provide a rapid overview of the question addressed in the study but still needs to be kept fairly modest since the image size cannot exceed 550x400 pixels.

Thank you again for giving us to consider your manuscript for EMBO Reports, I look forward to your minor revision.

REFEREE REPORTS

Referee #1:

The authors have addressed the points I have raised in a satisfying manner. In my opinion, the manuscript can be published in its current form.

Referee #2:

While the revisions by Kouznetsova et al. have eliminated all the minor problems, the criticized interpretations have all been retained without providing additional convincing justifications. Overall their efforts at improving the more problematic aspects of the original version have been minimal. This is unfortunate. As pointed out in the evaluation of the original version, the descriptive data of Kouznetsova et al. is certainly of great interest for researchers in the field of mouse oocyte meiosis.

1. There is probably general agreement that the accuracy of scoring kinetochore attachment types (in particular merotelic) by immunofluorescence with oocytes fixed after cold treatment is not perfect. Zhang et al. (2017) have also used this strategy and have found around 5fold fewer "erroneous" kinetochore-microtubule attachments at metaphase II than Kouznetsova et al. In their rebuttal, various methodological details are pointed out and the protocol by Vallot et al. (2018) which was used here is declared to be superior. Kouznetsova et al. indicate that in their hands the protocol used by Zhang et al. (2017) would have resulted in a detachment of a majority of kinetochores, while the cold treatment and fixation according to Vallot et al. (2018) entails the use of a microtubule-stabilizing buffer and finally higher resolution microscopy. Since Zhang et al. (2017) were able to score 84% of the attachments in control oocytes, it is impossible that their procedure necessarily results in detachment of a majority of kinetochores. Subtle details seem to be rather crucial. I do not see how Kouznetsova et al. can exclude that their assay conditions result in a greater survival of microtubules that are part of kinetochore microtubule bundles which however continue peripherally beyond the kinetochore rather than terminating end-on in the kinetochore. Mouse chromosomes are telocentric (while most human chromosomes are metacentric) and they might therefore have an increased frequency of such microtubules extending beyond the kinetochore and thereby giving the impression of a lateral attachment. Even with the increased microscopic resolution of Kouznetsova et al., it is impossible to exclude that some of the microtubules actually terminate end-on in a given kinetochore also in those attachments that were scored as "lateral". Kouznetsova et al. score as attachments as either "amphitelic", "syntelic", or "lateral/merotelic". The fact that "lateral/merotelic" are lumped together is presumably explained by the impossibility to resolve attachment types unequivocally. It appears that mouse mitotic cells have not yet been analyzed with the same approach. Overall therefore the basis of solid evidence for deciding that the fraction of erroneous kinetochore-microtubule attachments is unusually high in mouse oocytes at metaphase II seems frustratingly meagre. In general, publications that convincingly demonstrate merotelic attachments in cultured human cells reveal a pronounced stretching of the merotelically attached kinetochore. Stretching is not apparent in the examples shown by Kouznetsova et al. If a chromosome has a bi-polar attachment where a few microtubules extend peripherally beyond the kinetochore (as for example perhaps the case documented in Fig. 4E) instead of terminating end-

on, do we have to consider this to be an "erroneous" attachment? Normal segregation of a chromosome with such an "erroneous" attachment does not need any correction, neither before nor after anaphase onset. Around 50% of the chromosomes scored as "merotelic/lateral" (Fig. 2E, Fig. 4F) have additional properties (intercentromere distance, distance equator to plane) that are entirely within the range of those scored as "amphitelic". Could it be that many of these are not truly "erroneous" and hence do not undergo any "post-metaphase correction of erroneous kinetochore microtubule attachments (title)"? I do not see convincing evidence ruling out the possibility that Kouznetsova et al. overestimate the fraction of "erroneous" attachments perhaps quite drastically. Therefore, a careful discussion of the scoring difficulties and more cautious conclusions seem to be warranted.

2. Kouznetsova et al. insist on an absence of error correction before onset of anaphase II. Evidently, if a considerable fraction of the "erroneous" attachments are not truly erroneous, we do not have to be surprised by an absence of error correction in these cases. On the other hand, it is clearly evident that some of the chromosomes which produce anaphase II laggards display dramatic changes where the angle between intercentromere axis and spindle axis turns towards almost zero before anaphase II onset (Fig 6C). Kouznetsova et al. argue that since average intercentromere distance in the laggard-producing chromosomes does not change before anaphase onset there cannot be any error correction before anaphase II onset. This is not conclusive evidence ruling out that some selected chromosomes (perhaps those with truly erroneous attachments) undergo some error correction also before anaphase II onset. Maybe not all successfully. Nevertheless, what explains the plainly apparent rotation in some cases? Obviously, not all chromosomes are completely stable within the spindle. Also this issue deserves a more careful discussion and more cautious conclusions. This is not to say that the possibility of "post-metaphase correction" cannot be suggested.

3. As indicated in the original evaluation, the statistical basis for the conclusion that "less than 10% of the laggards missegregate and give rise to aneuploid gametes (abstract)" insufficient. But the revised version presents the "less than 10%" still as a conclusion in the abstract and in the discussion. While Kouznetsova et al. have found that less than 10% of the lagging chromatids (1 out of 13) that remain at anaphase II undergo nondisjunction and give rise to aneuploid gametes, they have not shown this (as this would require considerably more extensive analyses).

4. Lica et al. (1986) is cited in the revised version as a source for the intercentromere distance during metaphase in mitotically dividing mouse cells. However, Lica et al. (1986) analyse mitotic chromosomes isolated after 11 h in the presence of Colcemid. Obviously, these chromosomes are not normally bi-oriented within a mitotic spindle. The intercentromere distance cannot be expected to be normal (no stretch, but overcondensation). As the data from mitotically dividing somatic mouse cells does not seem to be available, it would have been great if Kouznetsova et al. would have added this. Citing Lica et al. (1986) as a source for intercentromere distance in normal mitotic metaphase in mouse cells is clearly inappropriate.

2nd Revision - authors' response

24 May 2019

Reviewer #2

1. There is probably general agreement that the accuracy of scoring kinetochore attachment types (in particular merotelic) by immunofluorescence with oocytes fixed after cold treatment is not perfect. Zhang et al. (2017) have also used this strategy and have found around 5fold fewer "erroneous" kinetochore-microtubule attachments at metaphase II than Kouznetsova et al. In their rebuttal, various methodological details are pointed out and the protocol by Vallot et al. (2018) which was used here is declared to be superior. Kouznetsova et al. indicate that in their hands the protocol used by Zhang et al. (2017) would have resulted in a detachment of a majority of kinetochores, while the cold treatment and fixation according to Vallot et al. (2018) entails the use of a microtubule-stabilizing buffer and finally higher resolution microscopy. Since Zhang et al. (2017) were able to score 84% of the attachments in control oocytes, it is impossible that their procedure necessarily results in detachment of a majority of kinetochores. Subtle details seem to be rather crucial. I do not see how Kouznetsova et al. can exclude that their assay conditions result in a greater survival of microtubules that are part of kinetochore microtubule bundles which however continue peripherally

beyond the kinetochore rather than terminating end-on in the kinetochore. Mouse chromosomes are telocentric (while most human chromosomes are metacentric) and they might therefore have an increased frequency of such microtubules extending beyond the kinetochore and thereby giving the impression of a lateral attachment. Even with the increased microscopic resolution of Kouznetsova et al., it is impossible to exclude that some of the microtubules actually terminate end-on in a given kinetochore also in those attachments that were scored as "lateral". Kouznetsova et al. score as attachments as either "amphitelic", "syntelic", or "lateral/merotelic". The fact that "lateral/merotelic" are lumped together is presumably explained by the impossibility to resolve attachment types unequivocally. It appears that mouse mitotic cells have not yet been analyzed with the same approach. Overall therefore the basis of solid evidence for deciding that the fraction of erroneous kinetochore-microtubule attachments is unusually high in mouse oocytes at metaphase II seems frustratingly meagre. In general, publications that convincingly demonstrate merotelic attachments in cultured human cells reveal a pronounced stretching of the merotelically attached kinetochore. Stretching is not apparent in the examples shown by Kouznetsova et al.

If a chromosome has a bi-polar attachment where a few microtubules extend peripherally beyond the kinetochore (as for example perhaps the case documented in Fig. 4E) instead of terminating end-on, do we have to consider this to be an "erroneous" attachment? Normal segregation of a chromosome with such an "erroneous" attachment does not need any correction, neither before nor after anaphase onset. Around 50% of the chromosomes scored as "merotelic/lateral" (Fig. 2E, Fig. 4F) have additional properties (intercentromere distance, distance equator to plane) that are entirely within the range of those scored as "amphitelic". Could it be that many of these are not truly "erroneous" and hence do not undergo any "post-metaphase correction of erroneous kinetochore microtubule attachments (title)"? I do not see convincing evidence ruling out the possibility that Kouznetsova et al. overestimate the fraction of "erroneous" attachments perhaps quite drastically. Therefore, a careful discussion of the scoring difficulties and more cautious conclusions seem to be warranted.

Answer to the reviewer:

The visualization protocol we have used (employing a microtubule-stabilizing buffer in combination with cold-treatment) is widely used for analyzing the MT-kinetochore attachments (see for example Greaney et al., 2018). We have made all possible efforts to make sure that the scoring method is as accurate as possible both by optimizing the protocol and also by using an improved microscopic resolution for image acquisition (see also our answer to the reviewer in our previous rebuttal letter).

The text book definition of amphitelic attachments are that these are "correct", and that others modes of attachments are "erroneous". We believe that introduction of a new definition: "truly erroneous" attachments, as suggested by the reviewer, would be confusing.

In order address the concerns of the reviewer and make our conclusions more cautious, we now have used expression "aberrant attachments" instead of "erroneous attachments" in the manuscript title and the manuscript text. We have revised the text in the Result section of the manuscript to clarify the methodology approach and possible scoring difficulties, as suggested by the reviewer. We have also included a reference to a methodology article published in *Methods in Molecular Biology* further describing the method and challenges involved (pages 9-10):

We next visualized kinetochore-MT attachments in fixed oocytes after release from CSF-mediated arrest and found that 78±4% of the chromosomes displayed amphitelic attachments, while the remaining chromosomes (3-6 per oocyte) had bi-directional merotelic/lateral attachments (Fig. 4 E-F, Fig. S4A). This is more than previously reported (about 5% of attachments were scored as merotelic or lateral in MII oocytes by [13]). The discrepancy between our result and those reported by [13] is most probably explained by the use of a different visualization protocols. We have used a protocol described in [16], where cells undergo cold treatment to remove less stable microtubules that are not attached to the kinetochores [22] in a stabilizing buffer instead of PBS as used by [13], to preserve MTs. In addition, we acquired the images with a microscope equipped with an AiryScan microscope module (Zeiss) to achieve super-resolution, allowing improved identification of thin MTs. Importantly, the percentage of merotelic/lateral attachments that we score before and after the CSF-mediated metaphase arrest release did not change.

We have also changed the abstract of the manuscript to ensure that the conclusions of our results are carefully formulated (page 2):

The accuracy of the two sequential meiotic divisions in oocytes is essential for creating a haploid gamete with a normal chromosomal content. Here, we have analysed the 3D dynamics of chromosomes during the second meiotic division in live mouse oocytes. We find that chromosomes form stable kinetochore-microtubule attachments at the end of prometaphase II stage that are retained until anaphase II onset. Remarkably, we observe that more than 20% of the kinetochore-microtubule attachments at the metaphase II stage are merotelic or lateral. However, less than 1% of all chromosomes at onset of anaphase II are found to lag at the spindle equator and less than 10% of the laggards missegregate and give rise to aneuploid gametes. Our results demonstrate that aberrant kinetochore-microtubule attachments are not corrected at the metaphase stage of the second meiotic division. Thus, the accuracy of the chromosome segregation process in mouse oocytes during meiosis II is ensured by an efficient correction process acting at the anaphase stage.

2. Kouznetsova et al. insist on an absence of error correction before onset of anaphase II. Evidently, if a considerable fraction of the "erroneous" attachments are not truly erroneous, we do not have to be surprised by an absence of error correction in these cases. On the other hand, it is clearly evident that some of the chromosomes which produce anaphase II laggards display dramatic changes where the angle between intercentromere axis and spindle axis turns towards almost zero before anaphase II onset (Fig 6C). Kouznetsova et al. argue that since average intercentromere distance in the laggard-producing chromosomes does not change before anaphase onset there cannot be any error correction before anaphase II onset. This is not conclusive evidence ruling out that some selected chromosomes (perhaps those with truly erroneous attachments) undergo some error correction also before anaphase II onset. Maybe not all successfully. Nevertheless, what explains the plainly apparent rotation in some cases? Obviously, not all chromosomes are completely stable within the spindle. Also this issue deserves a more careful discussion and more cautious conclusions. This is not to say that the possibility of "post-metaphase correction" cannot be suggested.

Answer to the reviewer:

As suggested by the reviewer we have introduced a more cautious interpretation of our results (page 12):

Though it cannot be excluded that the observed dynamic rotation of laggard-producing chromosomes is coupled to attachment error correction, the variability of their inter-centromere distances is similar to what is observed for normally segregating chromosomes (Fig. EV5H), indicating a lack of additional attachment correction activity for laggard-producing chromosomes when compared to the normally segregating chromosomes.

3. As indicated in the original evaluation, the statistical basis for the conclusion that "less than 10% of the laggards missegregate and give rise to aneuploid gametes (abstract)" insufficient. But the revised version presents the "less than 10%" still as a conclusion in the abstract and in the discussion. While Kouznetsova et al. have found that less than 10% of the lagging chromatids (1 out of 13) that remain at anaphase II undergo nondisjunction and give rise to aneuploid gametes, they have not shown this (as this would require considerably more extensive analyses).

Answer to the reviewer:

The number we give (less than 10%) contains only one significant digit (i.e. not 7.7%, instead "less than 10%"). In the text of the manuscript we have clearly indicated the number of analyzed chromatids (page 12 and page 14). We believe that it is reasonable to present this information with this cautious approach.

4. Lica et al. (1986) is cited in the revised version as a source for the intercentromere distance during metaphase in mitotically dividing mouse cells. However, Lica et al. (1986) analyse mitotic chromosomes isolated after 11 h in the presence of Colcemid. Obviously, these chromosomes are not normally bi-oriented within a mitotic spindle. The intercentromere distance cannot be expected to be normal (no stretch, but overcondensation). As the data from mitotically dividing somatic mouse cells does not seem to be available, it would have been great if Kouznetsova et al. would have added this. Citing Lica et al. (1986) as a source for intercentromere distance in normal mitotic metaphase in mouse cells is clearly inappropriate.

Answer to the reviewer:

We have as suggested by the reviewer removed the reference to Lica et al., 1986. We have instead introduced a more up to date reference to Ding et al., 2013 (page 9). In this article the authors measure intercentromere distance in mouse embryonic fibroblasts after brief treatment in

proteasome inhibitor MG132, a drug that does not affect kinetochore-microtubule attachments (Fig.3D in Ding et al., 2013).

3rd Editorial Decision

7 June 2019

Thank you for submitting your revised manuscript. I have now looked at everything and all looks fine. Therefore I am very pleased to accept your manuscript for publication in EMBO Reports.

Corresponding Author Name: Anna Kouznetsova

Manuscript Number: EMBOR-2019-47905